# KITE: Keypoint-Conditioned Policies
# for Semantic Manipulation

**Priya Sundaresan , Suneel Belkhale, Dorsa Sadigh, Jeannette Bohg**

Stanford University

**Abstract:** While natural language offers a convenient shared interface for humans and robots, enabling robots to interpret and follow language commands remains a longstanding challenge in manipulation. A crucial step to realizing a performant instruction-following robot is achieving *semantic manipulation* — where a robot interprets language at different specificities, from high-level instructions like 'Pick up the stuffed animal' to more detailed inputs like 'Grab the left ear of the elephant.' To tackle this, we propose KITE: Keypoints + Instructions to Execution, a two-step framework for semantic manipulation which attends to both *scene* semantics (distinguishing between different objects in a visual scene) and *object* semantics (precisely localizing different parts within an object instance). KITE first grounds an input instruction in a visual scene through 2D image keypoints, providing a highly accurate object-centric bias for downstream action inference. Provided an RGB-D scene observation, KITE then executes a learned keypoint-conditioned skill to carry out the instruction. The combined precision of keypoints and parameterized skills enables fine-grained manipulation with generalization to scene and object variations. Empirically, we demonstrate KITE in 3 real-world environments: long-horizon 6-DoF tabletop manipulation, semantic grasping, and a high-precision coffee-making task. In these settings, KITE achieves a $75\%$, $70\%$, and $71\%$ overall success rate for instruction-following, respectively. KITE outperforms frameworks that opt for pre-trained visual language models over keypoint-based grounding, or omit skills in favor of end-to-end visuomotor control, all while being trained from fewer or comparable amounts of demonstrations. Supplementary material, datasets, code, and videos can be found on our website.[1]

**Keywords:** Semantic Manipulation, Language Grounding, Keypoint Perception

## 1 Introduction

Language has the potential to serve as a powerful communication channel between humans and robots in homes, workplaces, and industrial settings. However, two primary challenges prevent today's robots from handling free-form language inputs. The first is enabling a robot to reason over *what* to manipulate. Instruction-following requires not only recognizing task-relevant objects from a visual scene, but possibly refining visual search to specific features on a particular object. For instance, telling a robot to "Open the top shelf" vs. "Yank open the bottom shelf" of a cabinet requires not only parsing and resolving any liberties taken with phrasing and localizing the cabinet in the scene (*scene* semantics), but also identifying the exact object feature that matters for the task — in this case the top or bottom handle (*object* semantics). In this work, we refer to instruction-following with scene and object awareness as *semantic manipulation*. Similarly, pick-and-place is a standard manipulation benchmark [1, 2, 3, 4], knowing how to pick up a stuffed animal by the ear versus leg, or a soap bottle by the dispenser versus side requires careful discernment. After identifying what to manipulate, the second challenge is determining *how* the robot can accomplish the desired behavior, i.e., low-level sensorimotor control. In many cases, low-level action execution requires planning in SE(3) with 6 degrees-of-freedom (DoF), such as reorienting the gripper sideways to grasp and pull open a drawer.

---

{priyasun, belkhale}@stanford.edu, dorsa@cs.stanford.edu, bohg@stanford.edu
[1] https://tinyurl.com/kite-site

7th Conference on Robot Learning (CoRL 2023), Atlanta, USA.

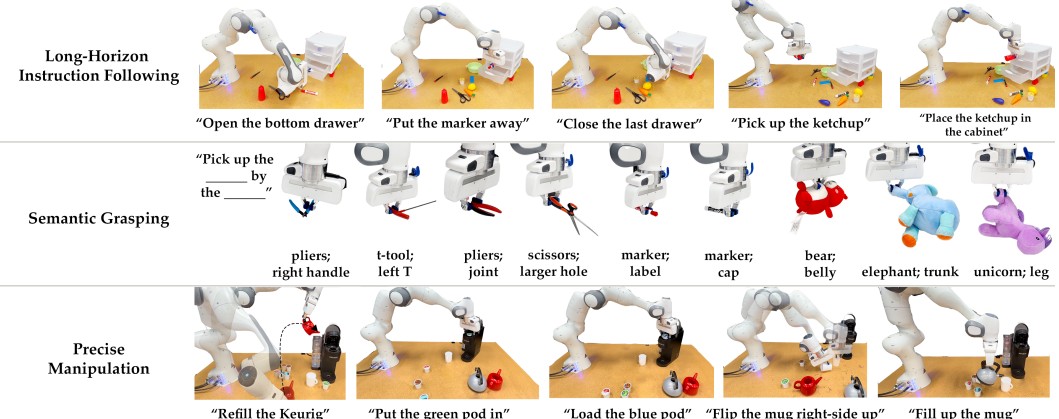

Figure 1: **Real-World Semantic Manipulation Environments:** We visualize our semantic manipulation framework KITE on three real-world environments: long-horizon instruction following, semantic grasping, and coffee-making. Using keypoint-based grounding, KITE contextualizes scene-level semantics ('Pick up the green/red/blue/brown coffee pod') as well as object-level semantics ('Pick up the unicorn by the leg/ear/tail', 'Open the cabinet by the top/middle/bottom shelf') and precisely executes keypoint-conditioned skills.

Going beyond grasping, we want robots to assist us in many daily real-world tasks which may require even finer-grained precision. Making coffee, for example, is a simple daily task for humans, but for a robot it involves complex steps like reorienting a mug from sideways to upright or carefully inserting a coffee pod into an espresso machine. Thus to achieve semantic manipulation, robots must extract scene and object semantics from input instructions and plan precise low-level actions accordingly.

Leveraging advances in open-vocabulary object detection [5, 6, 7, 8], prior works in language-based manipulation determinine *what* to manipulate via bounding boxes or keypoints obtained from pretrained [9] or fine-tuned vision language models (VLMs) [10]. So far, these works operate at the level of scene semantics (distinguishing amongst objects) rather than object semantics (identifying within-object features). In addition, these works do not apply VLMs to any complex manipulation beyond simple pick-and-place. To address these shortcomings, follow-up works couple the *what* and *how* subproblems together and learn end-to-end 6-DoF language-conditioned policies from demonstrations [11, 12, 13]. However, learning high dimensional action spaces from raw sensor inputs such as images or voxelized scene representations can require excessive amounts of data [12, 13] and can be difficult in high-precision tasks especially when using discretized actions [11].

Between approaches that leverage pre-trained visual representations from off-the-shelf VLMs, and those that plan directly from pixels or voxels, we lack an intermediate object-centric representation that can link natural language to scene and object semantics. Prior work has demonstrated that open-vocabulary VLMs can address scene semantics to some extent by locating different objects with coarse bounding boxes [9], but these representations are still too granular to precisely locate parts on objects. A suitable visual representation would be one that can represent both across-object or within-object features, and is interpretable enough to inform downstream 6-DoF action planning. We argue that keypoints provide this happy medium by offering a way to precisely pinpoint objects at the scene-level or even features within an object (*what*) and a way to condition downstream 6-DoF manipulation (*how*) on a region of interest.

In this work, we present **KITE:** Keypoints + Instructions To Execution, a flexible framework for semantic manipulation. KITE is decoupled into a grounding policy which maps input images and language commands to task-relevant keypoints, and an acting module which employs keypoint-conditioned skills to carry out low-level 6-DoF actions. We show that KITE can be trained from just a few hundred annotated examples for the grounding model, and less than 50 demos per skill for the acting module, while outperforming and generalizing better than methods that do not make use of either keypoints or skills. We experimentally evaluate KITE on semantic manipulation across three challenging real-world scenarios with varying tiers of difficulty: 6-DoF tabletop manipulation, semantic grasping, and coffee-making (Figure 1). Results indicate that KITE demonstrates fine-grained instruction following, while exhibiting a capacity for long-horizon reasoning and generalization.

## 2 Related Work

**Language-Based Manipulation**: Many recent works ground language to manipulation skills as a means for long-horizon instruction following. Several methods learn language conditioned policies end-to-end using imitation learning; however, end-to-end learning can require many demonstrations and can be brittle to novel scenes or objects [12, 11, 14, 15, 16, 17, 18, 19, 10, 20]. To improve sample efficiency, Shridhar et al. [11] predicts robot waypoints rather than low-level actions, conditioned on language and point cloud inputs. Waypoints alone do not specify *how* to go from one waypoint to another, and thus fail to capture dynamic tasks such as peg insertion or pouring motions. Other works take a hierarchical approach that first learn or define a library of language-conditioned skills, and then plan over skills with large language models (LLM) [21, 22, 23, 24, 25, 26]. However, each skill often requires hundreds of demonstrations. In addition, the LLM planner can only reason about the scene at a high level, lacking *visual grounding*. Alternatively, Liang et al. [27] query the LLM to generate code using an API for low-level skills, but predicting continuous parameters of these skills is challenging for an ungrounded LLM limiting this approach to tasks that do not require precision or dexterity. Vision-Language Models (VLM) are often proposed to ground LLM planners. For instance, Stone et al. [28] leverage a pretrained VLM to identify task-relevant objects from language. However, this approach has also been limited to pick and place tasks suggesting that today's pretrained VLMs struggle with more precise language instructions. In contrast, KITE maps language and vision directly to desired keypoints, enabling precise and semantic manipulation over long horizons.

**Skill-Based Manipulation**: Many prior works study multi-task manipulation by defining *skills* to represent sub-tasks, and then composing these skills over long horizons. These skills can either be learned to output each action or parameterized by expert-defined features. In reinforcement learning (RL), hierarchy can be imposed on the policy to learn both skills and composition end-to-end, but these methods can be sample inefficient and rarely integrate well with natural language [29, 30, 31]. Other RL works parameterize skills to reduce the action space size for sample efficiency, but these skills are usually rigid and cannot generalize to new settings [32, 33, 34]. Imitation learning (IL) aims to learn skills from demonstrations in a more sample efficient manner than RL, but these skills still fail to generalize to scene perturbations [35, 36]. Furthermore, in both IL and RL, connecting learned skills to precise language in a generalizable fashion is an open challenge. KITE avoids learning skills from scratch and instead defines a library of keypoint-conditioned skills, where the exact parameters of each skill are learned from demonstration. We show that keypoint-conditioned skills are sample efficient to learn and generalizable to new objects, while also easily integrated with precise language.

**Keypoints for Manipulation**: Keypoints have emerged in the literature as a more robust skill representation for manipulation [37, 10]. Keypoints are 2D points on images that serve as a natural intermediary between images and low-level behaviors. Several methods use keypoints to force the model to attend to the most important features in the input images [37]. Others predict keypoints and then translate keypoints into 3D points, or directly predict 3D points, to parameterize low level behaviors in a general and visually-grounded fashion [38, 10, 39, 40]. For example, Shridhar et al. [10] parameterize pick and place tasks using keypoints learned with image supervision, showing that this keypoint abstraction generalizes better to new objects. Keypoint action spaces have also helped in deformable object manipulation, for example in the domains of cloth folding, rope untangling, and food manipulation [41, 42, 43, 44, 45]. For many of these prior works, keypoints have yet to be integrated with language, and the methods that are linked to language are limited to a small library of primitives, usually focusing only on pick and place scenarios. Our approach defines a much broader library of keypoint-conditioned skills, and integrates keypoints with complex language instructions.

## 3 KITE: Keypoints + Instructions To Execution

In this work, our goal is to train an instruction-following agent capable of performing semantic manipulation at both scene and object-level granularity. We accomplish this with KITE, a sample-efficient and generalizable framework operating in two stages: *grounding* language into keypoints and *acting* on those keypoints. In this section, we first formalize the semantic manipulation problem (Section 3.1), discuss data collection (Section 3.2), and then discuss the training procedures for the grounding (Section 3.3) and acting modules (Section 3.4).

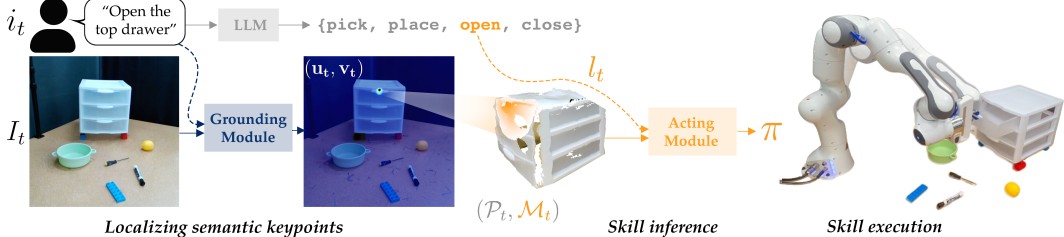

Figure 2: **KITE System Overview:** KITE receives an image observation $I_t$ along with user instruction $i_t$ and grounds these inputs to a 2D semantic keypoint in the image. After inferring which skill type $l_t$ is appropriate from a set of skill labels, KITE takes an RGB-D point cloud observation $\mathcal{P}_t$, annotated with the deprojected keypoint $\mathcal{M}_t$, and infers the appropriate waypoint policy $\pi$ for execution. After executing this action, KITE replans based on a new observation $(I_{t+1}, i_{t+1})$ and repeats the whole process.

### 3.1 Semantic Manipulation Problem Formulation

We aim to tackle instruction-following with scene and object-level semantic awareness using a library of *skills*. We assume each *skill* can be parameterized by 6-DoF waypoints, and we decouple each skill into a waypoint *policy* $\pi$ and *controller* $\rho$ to move between waypoints in a task-specific manner. Additionally, we assume each skill can be represented by a skill label $l$, e.g., pick, open, etc. We construct a library $\mathcal{L}$ of $M$ specialized skills where $\mathcal{L} = \{l^1 : (\pi^1, \rho^1), \ldots, l^M : (\pi^M, \rho^M)\}$ maps from a skill label to an underlying policy $\pi$ and controller $\rho$.

We assume access to multiple calibrated cameras that provide RGB and depth. We assume that at least one "grounding" camera can partially see all relevant objects. An observation $o_t = (I_t, \mathcal{P}_t)$ where $I_t \in \mathbb{R}^{W \times H \times 3}$ is the image from the grounding camera, and $\mathcal{P}_t \in \mathbb{R}^{D \times 6}$ is a multi-view point cloud. We denote low-level robot actions $a_t = (x, y, z, \psi, \theta, \phi)$ at time $t$ which consist of the end-effector position, yaw, pitch, and roll. We denote waypoints as $\kappa$ which is also a 6-DoF pose, but represents a high-level pose (e.g., grasp pose for pick) rather than a low-level action.

At time $t$, given an instruction $i_t$, we want to know *which* skill to execute by associating $i_t$ to a corresponding skill label $l_t \in \{l^1, \ldots, l^M\}$. Next, we aim to infer a 2D keypoint $[u, v]$ in the current visual observation $o_t$ which grounds $i_t$ to an associated object or object part. Finally, the chosen skill $(\pi, \rho) = \mathcal{L}[l_t]$ is executed (Figure 2). For each skill, we want to find a waypoint policy $\pi$ that takes as input the visual observation $o_t$ and 2D keypoint $[u_t, v_t]$ and outputs $K$ waypoints: $\pi : (o_t, [u_t, v_t]) \rightarrow \{\kappa^1, \ldots, \kappa^K\}$. Then, the associated controller $\rho$ can output a low-level trajectory between waypoints $\rho : \{\kappa^1, \ldots, \kappa^K\} \rightarrow \tau = \{(o_t, a_t), \ldots, (o_{t+T-1}, a_{t+T-1})\}$ (i.e. via linear interpolation, motion planning, etc.) for the robot to execute. For multi-step manipulation tasks, we restart the above process at each step with the new observation $o_{t+T}$ and paired language input $i_{t+T}$.

We consider instructions which refer to *scene* semantics, such as specifying desired spatial rearrangements of objects (e.g. "Pick up the lemon"), and *object* semantics, which reference desired object parts to be manipulated (e.g. "Grab the kangaroo stuffed animal by the tail"). As we do not assume access to the interaction history, our space of feasible language inputs excludes post-hoc feedback ("Pick up the *other* marker") or online-corrections ("No, to the *left*!"), which we leave to future work. Next we outline how KITE learns to predict keypoints (grounding) and learns each skill $\pi$ (acting).

### 3.2 Demonstration Collection

To learn both the grounding and acting modules, we collect a dataset $\mathcal{D}_\pi$ consisting of $N$ expert demonstrations per skill. Each demonstration has an initial observation, a list of $K$ waypoints, and a language instruction: $\mathcal{D}_\pi = \{(o_n, \{\kappa_n^1 \ldots \kappa_n^K\}, i_n) : n \in \{1, \ldots, N\}\}$. For instance, for a pick skill, we record the initial image and point cloud, provide an instruction (e.g., 'Pick up the lemon'), and then kinesthetically move the robot and record each end-effector waypoint. We use the calibrated robot-to-camera transformation to automatically project each robot end-effector pose $\kappa_n^j$ to 2D coordinates $[u_n, v_n]$ in the image plane of the camera used for grounding. For each skill, we train the acting module from $\mathcal{D}_\pi$. Aggregating across all skills yields a dataset of paired images, keypoint annotations, and language instructions with which to train the grounding module.

## 3.3 Grounding Module

The grounding module learns to identify 2D keypoints from RGB images that correspond to object features mentioned in an input instruction. We draw inspiration from recent works which use explicitly supervised [10, 38] or self-supervised keypoint attention networks [37, 46] to implement a grounding model $Q_{\text{ground}}$. Specifically, we learn a grounding function $Q_{\text{ground}}(u, v, I_t, i_t)$ representing the likelihood of keypoint $[u, v]$ given image $I_t$ and paired language instruction $i_t$. In this work, we attempt to learn a single-step look-ahead grounding function that takes a language input (e.g. "Put the lemon into the cabinet") and outputs the most immediately relevant keypoint (e.g. the pixel for the lemon if not already grasped, otherwise the pixel for the cabinet drawer to be placed) (see Fig. 2).

Given this grounding function $Q_{\text{ground}}$, we infer the 2D pixel in the image with highest likelihood:

$$[u_t, v_t] = \arg\max_{u,v} Q_{\text{ground}}(u, v, I_t, i_t) \tag{1}$$

In practice, we implement $Q_{\text{ground}}$ using the two-stream architecture from [10] which fuses pre-trained CLIP [47] embeddings of visual and textual features in a fully-convolutional network to output a heatmap of $Q_{\text{ground}}$. The grounding function is trained with a binary cross-entropy loss between the predicted heatmaps and 2D Gaussian heatmaps centered at the ground-truth pixel.

## 3.4 Acting Module

Although keypoints can pinpoint both scene and object semantics, they critically lack the 3D geometric context necessary to recover precise 6-DoF actions for a given task. For instance, the command "Pick up the bowl" may result in a predicted keypoint located at the bottom of a bowl, where there is no feasible grasp The exact 6-DoF actions are also dependent not just on the keypoint, but also language: "pick the lemon" and "cut the lemon" have similar keypoints but require completely different actions. We need a way to *refine* a predicted keypoint into candidate 6-DoF actions based on a desired language command, which we discuss next.

**Skill Selection:** Given a free-form language instruction, KITE first leverages the knowledge of LLMs to determine the appropriate skill label (e.g. $i_t =$"Put the lemon in the cabinet" should result in the LLM outputting $\hat{l}_t = $ 'pick_place'), following prior work [48, 21]. The procedure entails prompting the LLM, in our case OpenAI's text-davinci-003 [49], with in-context examples of instructions and the appropriate skill type (see Appendix B.3 for examples of our prompting strategy). At test-time, we concatenate the example prompt with instruction $i_t$ and generate skill label $\hat{l}_t \in \{l^1, \ldots, l^M\}$ using the LLM. Then, we obtain the skill, consisting of the waypoint policy $\pi$ and controller $\rho$ via lookup in the library: $(\pi, \rho) = \mathcal{L}[\hat{l}_t]$.

**Learning Waypoint Policies:** Given the keypoint $[\hat{u}_t, \hat{v}_t]$ predicted by the grounding module and skill label $\hat{l}_t$, we need to learn a waypoint policy $\pi$ to perform the skill. KITE learns $\pi$ for each skill from demonstrations of keypoints $\{\kappa^1, \ldots, \kappa^K\}$. The waypoint policy $\pi$ takes a point cloud $\mathcal{P}_t$ and keypoint $[\hat{u}_t, \hat{v}_t]$ as input, and aims to output $K$ waypoints $\{\kappa^1, \ldots, \kappa^K\}$ to execute the chosen skill.

In KITE we align both 3D point cloud and a 2D keypoint representations by "annotating" $\mathcal{P}_t$ with the keypoint. We do this by first taking the depth image $D_t$ from the same view as $I_t$, and deprojecting all nearby pixels within a radius $R$: $\mathcal{K}_R = \{[u, v] \in I_t, \|[u, v] - [\hat{u}_t, \hat{v}_t]\| < R\}$ to their associated 3D points $\mathcal{P}_R = \{(x, y, z) = \texttt{deproject}(u, v), \forall (u, v) \in \mathcal{K}_R\}$. This yields a set of "candidate" points to consider for interaction. In the bowl grasping example, $\mathcal{P}_R$ would be points on the bottom of the bowl. Next, we augment the point cloud $\mathcal{P}_t$ with a 1-channel mask $\mathcal{M}_t \in \mathbb{R}^{N \times 1}$ (Fig. 2). For any point $(x, y, z) \in \mathcal{P}_t$, the mask channel label is 1 if $(x, y, z) \in \mathcal{P}_R$ (i.e., the point is in close proximity to the deprojected keypoint) and 0 otherwise.

Given the pointcloud and keypoint mask, KITE predicts all $K$ waypoints *relative* to individual points in the point cloud. For each point, we classify which of the $K$ waypoints it is nearest to, along with the offset to the desired 7-DoF end-effector pose (position and quaternion) for each of the $K$ waypoints. To do so, we adapt the PointNet++ [50] architecture, and define $Q_\pi : (\mathcal{P}_t, \mathcal{M}_t) \to \mathcal{P}_\pi$ where $\mathcal{P}_\pi \in \mathbb{R}^{N \times d}$ and $d = K \times (1 + 3 + 4)$. Continuing the example of grasping a bowl, the predicted pose offsets for each point on the bottom of the bowl (ungraspable) should lead to the bowl rim (graspable). See Appendix A for more details about the actor. We supervise $Q_\pi$ using the

following per-point loss:

$$L_{\text{skill}} = \lambda_{\text{cls}} CE(\hat{k}, k) + \lambda_{\text{ori}}(1 - \langle \hat{q}_{\hat{k}}, q_k \rangle) + \lambda_{\text{pos}} L1([\hat{x}_{\hat{k}}, \hat{y}_{\hat{k}}, \hat{z}_{\hat{k}}], [x_k, y_k, z_k]) \quad (2)$$

The first term corresponds to the 1-hot cross-entropy classification loss between the predicted waypoint index $\hat{k}$ and the true nearest waypoint index $k$. The remaining terms supervise the predicted gripper orientation and position using waypoint $\kappa^k$ for only the points that have classification label $k$, so as to only penalize points that matter (are in close proximity) to the $\kappa^k$.

**Action Module Inference:** At test time, given a point cloud $\mathcal{P}_t$ with associated keypoint mask $\mathcal{M}_t$, we use $Q_\pi$ to obtain $\hat{\mathcal{P}}_\pi$. By taking the highest likelihood point for each of the $K$ indices (representing the points nearest each of the $K$ waypoints). Then, we index the predicted end-effector poses in $\hat{\mathcal{P}}_\pi$ by these $K$ indices, resulting in $K$ waypoints $\{\hat{\kappa}^1, \ldots, \hat{\kappa}^K\}$. Finally, we obtain the final trajectory to carry out the skill with using the skill-specific controller: $\tau = \rho(\{\hat{\kappa}^1, \ldots, \hat{\kappa}^K\})$.

In summary, KITE's full pipeline first grounds a language command $i_t$ in an observation $o_t$ via $Q_{\text{ground}}$ to infer keypoints (Section 3.3), infers the skill label $l_t$ (Section 3.4), maps the label to a skill and controller $(\pi, \rho) = \mathcal{L}[l_t]$, then and executes $\pi$ and $\rho$, and finally replans.

## 4 Experiments

In this section, we aim to answer the following questions: (Q1) How well does KITE handle scene semantic instructions? (Q2) How well does KITE handle precise object-semantic instructions? (Q3) Does KITE's scene and object semantic awareness generalize to unseen object instances? and (Q4) Can KITE's primitives capture precise and dexterous motions beyond pick and place? We first outline KITE's key implementation details and the baseline methods we benchmark against. Finally, we analyze KITE's comparative and overall performance across three real-world environments which stress test scene and object-aware semantic manipulation (Section 4.1).

**Implementation Details:** Across all evaluation environments, we specify a library of skills and collect 50 kinesthetic demonstrations per-skill. In order to improve precision of the grounding module, and because keypoint supervision is easy to obtain compared to kinesthetic teaching, we supplement the grounding dataset obtained from kinesthetic data collection by manually labeling a small amount of images with paired language instructions (0.75:1 supplemental to original samples ratio). We implement the grounding module according to the architecture from [10] and each waypoint policy in the acting module as a PointNet++ backbone [50] with a point cloud resolution of 20K points. See Appendix A for more details and visualizations of grounding model predictions.

**Baselines:** We benchmark KITE's performance against two state-of-the-art instruction-following frameworks. The first is PerAct [11], which trains a PerceiverIO [51] transformer backbone to predict waypoint poses end-to-end conditioned on a voxelized scene representation and language. In comparing against PerAct, we hope to understand whether KITE's use of keypoint-parameterized skills can offer better precision over end-to-end actions. To understand the value of keypoint-based grounding over frozen representations obtained from VLMs, we compare to RobotMoo [9], which extends a library of language-conditioned skills to additionally condition on segmentation masks from an open-vocabulary object detector. Since exact models and data were not released, we use a state-of-the-art VLM and our set of learn skills for RobotMoo. See Appendix A.3 for more details.

### 4.1 Real-World Evaluation

We explore three real-world manipulation environments that provide a rich testbed to explore KITE's sensitivity to scene and object semantics. Task variations are detailed in Appendix B.1. Across all experimental trials, we use a Franka Emika 7DoF robot and 3 Realsense D435 RGB-D cameras.

**Tabletop Instruction-Following:** We train a library of four skills: {pick, place, open, close} to reorganize a tabletop environment with 15 different household objects and an articulated storage organizer with three pull-out drawers (see Figure 1, Appendix B.1). While we do not test on completely unseen objects, we randomly vary the positions of objects on the table and the degree of clutter by adding distractor objects to the scene.

Table 1 compares KITE against PerAct and RobotMoo in this setting. We evaluate all approaches with 12 trials of instruction-following across three tiers of difficulty, ranging from a few objects on

the table and fairly straightforward language instructions (Tier 1), a visually cluttered table (Tier 2), and a cluttered table with more ambiguous instructions (Tier 3). See Appendix B.1 for examples of the objects considered and variations across tiers.

We first evaluate individual actions (`open`, `close`, `pick`), finding KITE to be the most robust and repeatable. KITE's use of precise keypoint grounding enables scene semantic awareness (Q1) over different objects (`pick, place`) and object semantic understanding (Q2) by distinguishing amongst different drawer handles with the `open` and `close` skills. PerAct's disadvantage is its discrete visual space, where any slight 1-off voxel predictions can make it difficult to grasp objects or cabinet handles. Due to the weak classification objectives it is trained on, its most common failure mode is misclassified gripper opening/closing actions. These failures are alleviated by the parameterized skills used in KITE and RobotMoo. Unsurprisingly, RobotMoo does well at grasping different objects referenced in language, as VLMs trained on internet-scale data have strong object priors.

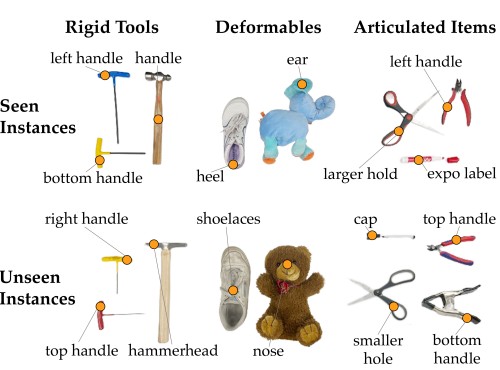

Figure 3: **Semantic Grasping Experimental Setup:** We evaluate KITE on semantic grasping across rigid tools, deformable objects, and articulated items. We show 17 of the 20 objects tested along with ground-truth semantic labels for different features. The top row includes objects seen during grounding module training, and the bottom consists of unseen object instances.

Still, RobotMoo struggles with object semantics like the distinction amongst top, middle, or bottom drawers when opening or closing. We find that KITE is also the most competitive framework for long-horizon sequential reasoning (last two columns in Table 1), and the most common failures still include grasping the wrong object or with slightly misaligned gripper poses. RobotMoo's inability to reason over multiple cabinet handles for opening and closing impedes its long-horizon performance, whereas PerAct's compounding precision errors render long horizon tasks especially difficult.

|        |          | open | close | pick | pick→place | open→pick→place→close |
|--------|----------|------|-------|------|-----------|-----------------------|
| Tier 1 | KITE     | **1**    | **0.92**  | **0.83** | **0.75**      | **0.75**                  |
|        | RobotMoo | 0.33 | 0.41  | 0.75 | 0.41      | 0.08                  |
|        | PerAct   | 0.08 | 0.5   | 0.33 | 0.08      | N/A                   |
| Tier 2 | KITE     | **1**    | **0.83**  | **0.76** | **0.66**      | **0.42**                  |
|        | RobotMoo | 0.36 | 0.36  | 0.55 | 0.36      | 0.09                  |
| Tier 3 | KITE     | **0.75** | **0.83**  | **0.58** | **0.66**      | **0.58**                  |
|        | RobotMoo | 0.33 | 0.42  | 0.5  | 0.42      | 0.0                   |

Table 1: **Tabletop Instruction Following Results:** Across 12 trials per method per tier, KITE outperforms both RobotMoo and PerAct for individual actions (`open`, `close`, `pick`) and chaining together up to four actions in sequence. We test all approaches on Tier 1 (fewer objects, straightforward language), Tier 2 (more objects, straightforward language), and Tier 3 (more objects, more free-form language). KITE's use of parameterized skills gives it an edge with precision over PerAct, which is highly susceptible to one-off voxel predictions. This makes skills like opening and picking especially hard, and renders the approach virtually ineffective for higher complexity tiers (2 and 3). RobotMoo is the most competitive approach to KITE, but its main pitfall is a lack of object semantic awareness such as distinguishing amongst different-level cabinet handles.

**Semantic Grasping:** Aside from recognizing and manipulating different objects, we explore in greater detail whether KITE can perform *object*-semantic manipulation (Q2). We evaluate KITE on the task of *semantic grasping*, with instructions of the form "Pick up the X by the Y" (i.e. 'stuffed bear' and 'ear'; 'marker' and 'cap'; 'shoe' and 'laces') (examples in Fig. 1). For these trials, we train $Q_{\text{ground}}$ on a subset of rigid tools, deformable items, and articulated items (Fig. 3) and retain the keypoint-conditioned `pick` skill from Section 4.1. We summarize the findings in Table 2 with 26 trials per category of items, noting that KITE can achieve precise semantic grasping with generalization to unseen object instances (Q2, Q3). We omit a comparison to PerAct as its difficulties

with pick-and-place in the tabletop environment are only exacerbated in the semantic grasping setting where specific intra-object features matter. In the trials summarized in Table 2, KITE outperforms RobotMoo, suggesting the utility of keypoints to pinpoint specific object parts compared to coarse segmentation masks or bounding boxes output by VLMs. We also observe that the majority of KITE's failures in this setting are due to misinterpretations with symmetry (i.e. grasping the left instead of right handle of the pliers), rather than a completely erroneous keypoint as is common in RobotMoo. We posit that this could be alleviated with more diverse data of object semantic variations.

| | | Rigid Tools | Deformable Objects | Articulated Items | Failures A | B | C |
|---|---|---|---|---|---|---|---|
| Seen Instances | KITE | **0.77** | **0.77** | **0.70** | 5 | 3 | 3 |
| Unseen Instances | KITE | **0.70** | **0.54** | **0.70** | 4 | 5 | 4 |
| All | RobotMoo | 0.23 | 0.35 | 0.19 | 2 | 36 | 7 |

Table 2: **Semantic Grasping Results**: Across 20 total objects, 3 diverse object categories, and 26 trials per method per category, KITE achieves the highest rate of pick success for various object semantic features (Fig. 3), and with the least severity of failures. We categorize failure modes as follows, with (A) denoting a symmetry error (picking the left instead of right handle), (B) representing a grounding error with an erroneous keypoint prediction, and (C) indicating a manipulation failure (wrong inferred orientation or slip during grasping).

**Coffee-Making:** Finally, we seek to answer whether KITE can execute fine-grained behaviors from instructions (Q4) by studying a coffee-making

| | reorient_mug | pour_cup | refill_keurig | load_pod |
|---|---|---|---|---|
| KITE | 8/12 | 9/12 | 8/12 | 9/12 |

Table 3: **Coffee-Making Results:** KITE handles fine-grained manipulation across 4 skills requiring highly precise manipulation.

scenario with four skills: {reorient_mug, pour_cup, refill_keurig, load_pod} (examples in Fig. 1). We evaluate on the same object instances seen in training, but subject to spatial variations and language variations (i.e. 'Place the blue/red/green/brown pod in the machine', 'Pour the red/grey pitcher into the mug/Keurig refill area,'Place the cup/mug that's sideways right-side-up.'). Even for these very fine-grained motions, KITE is able to follow instructions with 67-75% success (Table 3). The main failures reside with low-level control errors rather than grounding, such as partial coffee pod insertion, misaligned mugs and pitchers during pouring, or slippage during mug reorientation. This suggests that the individual skills could benefit from scaling up demonstration collection, while retaining the existing grounding modules.

## 5 Discussion

**Summary** In this work, we present KITE a framework for semantic manipulation that identifies 2D task-relevant keypoints and extrapolates 6-DoF actions accordingly. By leveraging 2D keypoints to precisely localize semantic concepts, KITE is adept at recognizing semantic labels both across different object instances and on different regions within the same object. KITE does action-planning by drawing from a library of parameterized skills. Empirically, we find that KITE surpasses existing language-based manipulation frameworks along the axes of scene semantic awareness and object semantic awareness. We also find that KITE can be trained from orders of magnitude less data and with large precision gains over end-to-end approaches, while exhibiting an ability to generalize and operate over extended horizons. Finally, we show that KITE offers a flexible interface for instruction-following, including tabletop rearrangement, fine-grained grasping, and dexterous manipulation.

**Limitations and Future Work** One limiting factor of KITE is its reliance on building a library of skills. However, we show that a relatively small library of keypoint-parameterized skills is expressive enough to accomplish many standard manipulation tasks with object variations over an extended horizon. Additionally, KITE requires less than 50 demonstrations per new skill, meaning that adding new skills is fairly straightforward. We also note that KITE's grounding module is trained from scratch. As VLMs continually improve and in the future may be able to pinpoint keypoints in images, it would be interesting to replace or enhance KITE's grounding module with these models. Additionally, we acknowledge that KITE currently executes skills in an open-loop manner as parameterized by waypoints. In, the future, we are excited to extend KITE's skills with closed-loop feedback and extend the complexity of these skills to even more dexterous settings.

## Acknowledgments

This work is in part supported by funds from NSF Awards 2132847, 2006388, 2218760, as well as the Office of Naval Research, and FANUC. Toyota Research Institute provided funds to support this work. We also thank Sidd Karamcheti and Yuchen Cui for their helpful feedback and suggestions. Priya Sundaresan is supported by an NSF GRFP.

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

# KITE: Keypoints + Instructions To Execution
# Supplementary Material

In this section, we outline additional details regarding the implementation of KITE, the real-world environments studied, and qualitative results of all methods. Please refer to our website to best understand the task diversity and qualitative performance through videos of KITE performing real-world semantic manipulation.

## A    Implementation Details

### A.1    KITE

As discussed in Section 3.4, KITE trains a PointNet++ model to output all $K$ relative waypoints and a one-hot waypoint index for each point in the point cloud. The auxiliary one-hot classification output layer classifies *which* waypoint each point in the point cloud is most relevant for manipuation (nearest to). In practice, many skills like `pick` or `place` can be parameterized by just $K = 1$ waypoint (where to grasp, where to place). In this case, the one-hot classification output is reduced to binary classification of which points are near the graspable object or target location, respectively. For more general skills parameterized by $K$ waypoints, we can supervise the $K$-th end-effector pose predictions per-point by taking the loss of the predicted and ground truth gripper pose for that point compared to ground truth. This loss is provided in the main text in Eq. (2).

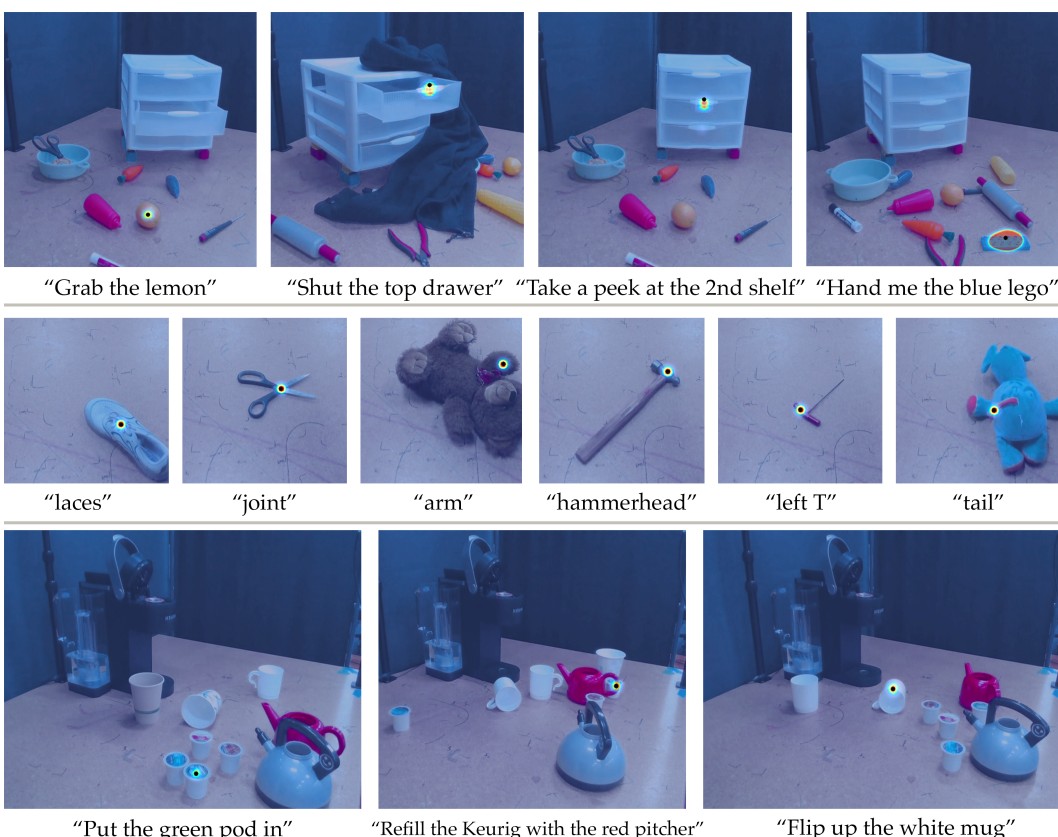

"Grab the lemon"          "Shut the top drawer"  "Take a peek at the 2nd shelf"  "Hand me the blue lego"

"laces"          "joint"          "arm"          "hammerhead"          "left T"          "tail"

"Put the green pod in"          "Refill the Keurig with the red pitcher"          "Flip up the white mug"

Figure 4: **KITE Grounding Predictions:** KITE's grounding model is able to accurately predict keypoints for both scene semantic instructions (e.g., "grab the lemon" and "put the green pod in") and object semantic instructions (e.g., "shut the top drawer" and "take a peek at the 2nd shelf".

To artificially scale the data KITE's grounding module is trained on, we apply various random colorspace and affine transformations to the dataset collected with all skills to augment 8X before training. We train the grounding module and each skill policy using the Adam optimizer with learning rate 0.0001, which takes 3 hours and 1 hour on an NVIDIA GeForce GTX 1070 GPU, respectively.

### A.2    PerAct

For each evaluation environment, we consolidate each of the skill datasets used to train KITE into one multi-task dataset with which to train PerAct. The input to PerAct is a $75^3$ voxel grid (although

the original PerAct implementation used a $100^3$ voxel resolution, we adjust our workspace bounds accordingly to retain the same voxel resolution). We represent waypoints as 1-hot encodings in this voxel grid, end-effector orientations as a discrete prediction over 5 bins each for yaw, pitch, and roll, and the gripper open/closed state as a binary indicator variable as in [11]. For each environment, we train PerAct for 7200 iterations.

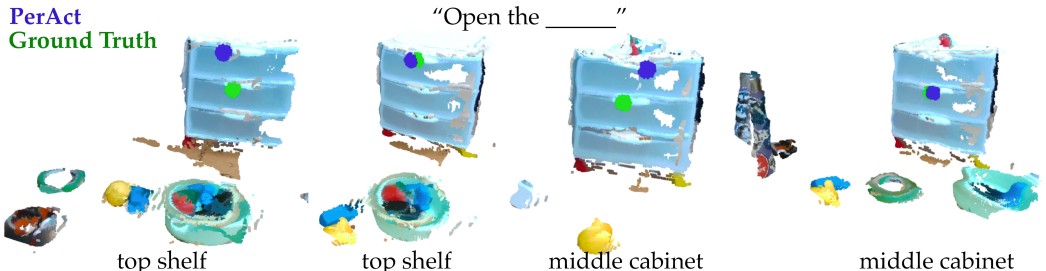

Figure 5: **PerAct Predictions:** We visualize PerAct predictions on the task of opening a cabinet with multiple drawers. Although PerAct exhibits some reasonable predictions (last column), it struggles with localizing the correct handle (1st, 3rd columns). Even when localizing the correct handle (2nd column), the slight imprecision of the predict vs. ground truth action can lead to downstream manipulation failure.

### A.3   RobotMoo

We note that the original implementation of RobotMoo leveraged the RT-1 [13] skill learning framework. This set of skills were trained with months of data collection, amassing thousands of trajectories for 16 object categories, and RobotMoo further extended these policies to 90 diverse object categories. As this is not reproducible in our setting, we implement RobotMoo by using KITE's library of skills, but conditioning them on VLM predictions instead of our keypoints. Specifically, while the original RobotMoo implementation used OwLViT, we use the more recent state-of-the-art open vocabulary object detectors Grounding DINO [8] and Segment Anything [52] jointly. With these models, we obtain segmentation masks for objects referenced in an input instruction. We take the center pixel coordinate of these segmentation masks as input to our acting module rather the output of $Q_{\text{ground}}$.

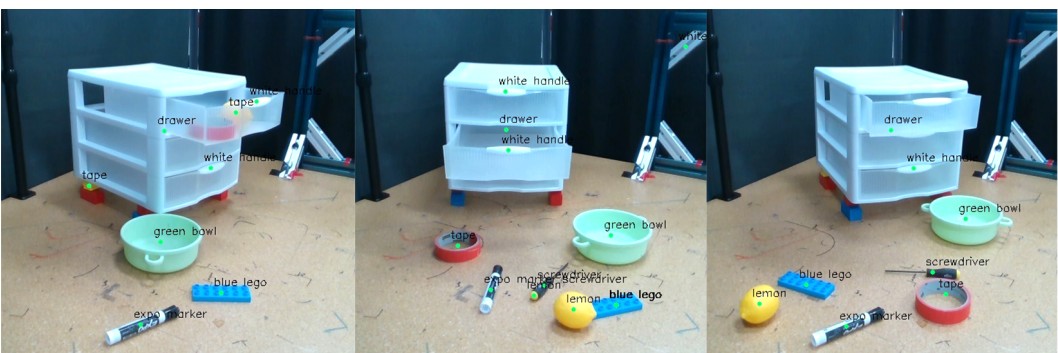

Figure 6: **RobotMoo Predictions:** We visualize the predictions for RobotMoo's perception stack on tabletop instruction following images. Although RobotMoo exhibits decent scene awareness and an ability to localize different object instances, it struggles with object semantics. Specifically, RobotMoo struggles to localize all the different drawers in the scene, let alone distinguish amongst the *top* vs. *middle* vs. *bottom* handles.

## B   Real-World Experimental Details

### B.1   Task Variations

For each real environment used in evaluation, we stress-test all methods across task variations ranging from diversity in the input language instructions to amount of clutter and distractor objects. We summarize each axis of variation for long-horizon tabletop instruction following (Table 4), semantic grasping (Table 5), and coffee-making (Table 6) below.

| Tier | Skill | Variations | |
|---|---|---|---|
| | | **Language** | **Scene** |
| 1 | open | 'Open the [top/middle/bottom] cabinet' | randomized position of cabinet |
| | close | 'Close the [top/middle/bottom] drawer' | randomized position of cabinet |
| | pick | 'Pick up the [lemon/screwdriver/lego/bowl/expo marker]' | randomized object positions |
| | place | 'Put the [...] [away, in the [...] drawer, in the bowl]' | randomized object positions |
| 1 | open | 'Open the [top/middle/bottom] cabinet' | randomized position + distractor objects (clothes strewn) |
| | close | 'Close the [top/middle/bottom] drawer' | randomized position + distractor objects (clothes strewn) |
| | pick | 'Pick up the [lemon/screwdriver/lego/bowl/expo marker, eggplant, carrot, corn, lime, scissors, ketchup, coffee pod]' | randomized position + scene clutter |
| | place | 'Put the [...] [away, in the [...] drawer, in the bowl]' | randomized object positions + scene clutter |
| 3 | open | 'Yank open the top drawer' 'Give the 2nd drawer a tug' 'Take a peek at the 3rd drawer pls' 'Can you check the top drawer?' | randomized position + distractor objects (clothes strewn) |
| | close | 'Close it' 'Give it a push' 'Let's shut the drawer' 'Go ahead and close the top drawer' 'Close any open drawer' 'The one 3rd from the bottom needs to be shut' | distractor objects (clothes strewn) |
| | pick | 'Grab me the [...]' 'Do me a favor and get me the [...]' 'Can you pass me the [...]?' 'Get the [...]' 'Locate the [...]' 'Could you hand me the [...] please' | randomized object positions + scene clutter |
| | place | 'Grab the [...] and put it [...]' 'Take the [...] and place it [...]' 'Pick up the [...] and put it [...]' 'Fetch the [...] and drop it [...]' 'Plop the [...] into the bowl' | randomized object positions + scene clutter |

Table 4: **Tabletop Instruction Following Environment Variations**

| Category | Object | Language Variations |
|---|---|---|
| Rigid Tools | hammer | middle, end, tooltip, hammerhead, metal, wooden handle, center, tip |
| | T-tool | left side, left T, right side, right T, bottom handle, top handle |
| Deformable Objects | shoe | heel, toe, back, front, shoelaces, laces, lace-up area |
| | stuffed animal | head, nose, ear, tail, belly, foot, arm, leg, tummy, elephant trunk |
| Articulated Items | pliers | joint, left handle, right handle, top handle, bottom handle |
| | clamp | joint, left handle, right handle, top handle, bottom handle |
| | scissors | joint, left hole, right hole, smaller hold, bigger hold, larger hold |
| | marker + twist-off cap | cap, expo label, center, label, end |

Table 5: **Semantic Grasping Environment Variations**

## B.2 Primitive Instantiations

In this section, we describe the instantiation of our library of skills for each real-world environment:

**Tabletop Instruction Following:** We parameterize each skill in the tabletop manipulation setting via a single waypoint $\kappa = (x, y, z, \psi, \theta, \phi)$ specifying the primary point of interaction.

- open: With its gripper open and predicted orientation $(\psi, \theta, \phi)$, the robot approaches 5cm. away from a closed drawer handle at predicted position $(x, y, z)$. Next, the robot moves to $(x, y, z)$ in the same orientation and closes the gripper to grasp the cabinet handle. Finally, it executes a linear pull by moving to the approach position, keeping the orientation fixed, before releasing the handle.

- close: The robot approaches 5cm. away from an opened drawer handle at position $(x, y, z)$ with orientation $(\psi, \theta, \phi)$, then executes a linear push towards $(x, y, z)$ to close the drawer.

- pick: The robot approaches the object located at $(x, y, z)$, closes its gripper, and lifts 5cm.

- place: While holding an object grasped with the pick primitive, the robot moves 5cm above the desired place location $(x, y, z)$ with orientation $(\psi, \theta, \phi)$ and opens the gripper to its maximum width, releasing the object.

**Coffee-Making** In the coffee-masking tasks, we implement a library of 4 skills which test KITE's ability to handle precise or dynamic movements. Since `pour_cup`, `refill_keurig`, and `load_pod` all involve grasping, we finetune the `pick` skill from tabletop instruction-following with 50 demonstrations across pitchers and coffee pods, respectively. Then, we can parameterize each skill with a single waypoint $\kappa = (x, y, z, \psi, \theta, \phi)$ as follows:

- `reorient_mug`: The robot attempts to grasp a mug, initially oriented sideways, with pose $\kappa$ before resetting to a canonical upright (untilted) end-effector pose.
- `pour_cup` / `refill_keurig`: After grasping a pitcher, the robot moves to position $(x, y, z)$ denoting the position of the vessel to be poured into (cup or refill compartment of Keurig). Starting from an untilted end-effector pose, the robot gradually rotates at a constant velocity to $(\psi, \theta, \phi)$, denoting the final pour orientation.
- `load_pod`: After grasping a coffee pod with the `pick` primitive, the robot moves 2 cm. above $(x, y, z)$, the sensed position of the K-cup slot with orientation $(\psi, \theta, \phi)$. Next, the robot releases its grasp to drop the pod into the compartment. As this task requires high precision, it is often the case that after releasing the pod, it is not completely inserted or properly aligned. Thus, the `load_pod` primitive moves downward an additional 2cm in attempt to push the pod into place. We note that we do not evaluate this skill with real liquids for safety reasons, but measure success in terms of visual alignment between the pitcher and vessel.

| Skill | Language | Scene |
|---|---|---|
| reorient_mug | 'Flip the mug right-side up' 
 'Put the mug upright' 
 'Grab the mug and put it it right-side-up' 
 'Can you place the mug right side up?' 
 'Get the mug that's laying flat 
 and flip it upright' | randomized mug position, roll $(-\pi/2, \pi/2)$ |
| pour_cup | 'Fill up the mug that's right-side up' 
 'Pour me a glass' 
 'Pour the red pitcher into the mug' 
 'Grab the silver-handle pitcher and fill 
 up the brown cup' 
 'Refill the Dixie cup with the red pitcher' | randomized pitcher (red / gray) 
 + cups (compostable, Dixie, mug) |
| refill_keurig | 'Refill the espresso machine' 
 'Grab the red/silver pitcher and 
 'fill up the water compartment' | randomized pitcher (red / gray) 
 + randomized Keurig pose |
| load_pod | 'Load the blue K-cup' 
 'Can you put the red pod in?' 
 'Insert the green pod' 
 'Start a brew with the brown pod' | randomized coffee pod (red/blue/green/brown) 
 + randomized Keurig |

Table 6: **Coffee-Making Variations**

## B.3 LLM Prompting

**Skill Label Inference:** In this section, we briefly outline how KITE retrieves the skill label $l_t$ for input instruction $i_t$ via LLMs. Below, we provide a sample prompt which we feed as input to `text-davinci-003` to obtain $l_t$ in tabletop instruction following setting.

Listing 1: LLM Prompting for Skill Label Inference

```
i_t = input("Enter instruction:")
"""
Input: "Pick up the lemon"
Output: ["pick"]

Input: "Put the screwdriver away"
Output: ["pick", "place"]

Input: "Pls grab me the screwdriver and put it away"
Output: ["pick", "place"]

Input: "Grab the green bowl"
Output: ["grasp"]

Input: "Put the lemon in the bowl"
```

```
16  Output: ["pick", "place"]
17
18  Input: "Open the top drawer"
19  Output: ["open"]
20
21  Input: "Pls shut the drawer"
22  Output: ["close"]
23
24  Input: "put the expo marker away"
25  Output: ["pick", "place"]
26
27  Input: "put the Blue lego in the cabinet"
28  Output: ["pick", "place"]
29
30  Input: '%s'
31  Output:
32  """% i_t
```