# OpenReview forum: "KITE: Keypoint-Conditioned Policies for Semantic Manipulation"
_robot-learning.org/CoRL/2023/Conference — CoRL 2023 Poster_

### Official Review · Reviewer_C4rM · 2023-07-17

**Confidence:** 4
**Originality:** Good
**Technical Quality:** Good
**Clarity Of Presentation:** Good
**Impact:** 3

**Recommendation:**

Weak Accept: I recommend accepting the paper, but will not argue for my recommendation if the majority of other reviewers have a different opinion.

**Review:**

**Summary of strengths and weaknesses**

Strengths:

- The real demonstrations are impressive and compelling. Any pipeline that enables the diverse kinds of capabilities shown in the paper (semantic grasping, multi-step tabletop manipulation) directly from language + RGB-D on real-world objects is noteworthy.
- The paper showcases an overall system implementation that works and that is (mostly) described in sufficient detail to understand how to reproduce and build upon it.
- The main novel technical components (the locally conditioned "waypoint-pose predictor"/skill policy) make sense, both in their design and in the way they're integrated into the overall pipeline.

Weaknesses:


1. In my view, beyond the good overall system engineering, the main technical innovation is the local "waypoint pose" prediction. Thus, it’s a critical part of the paper to understand, but as it's written, I found it to be the most difficult part of the paper to understand (at least in terms of technical details). Some more detailed comments are mentioned with regard to this in the "clarity" section below.
2. A related weakness is the framing of the paper and the difficulty in knowing what the main takeaway messages are. The results are good and the gist of the pipeline is not difficult to understand, but there are so many terms and claimed advantages (*long horizon reasoning*, *generalization*, *instruction following*, *parameterized skills are better than end-to-end*, *need for both scene- and object-level semantic*) that its hard to know what I learned as a reader of the paper.
    - If we are to break it down a bit, one perspective (that I had as a reader) is below. Take it for what it's worth, but the general suggestion is to find a way to make it easier for a reader to walk away from the paper with a more clear "take-home message" in mind.
        - The main new technical component is the local skill policies - predicting a sequence of waypoint poses from a local point cloud, parameterizing these as offsets from the point cloud points, using a different pose prediction network for each skill, and training these from a modest # of demonstrations.
        - The motivation for building these local skill policies is to utilize them within an "RGB-D + language -> 6-DoF EE trajectory" instruction-following manipulation framework. By doing so, this will expand upon the capabilities of previous instances of "manipulation from language instructions" pipelines, so that (i) the executed motions can be more diverse, expressive and accurate, (ii) they can integrate with more kinds of instructions (specifying a desired part to interact with), and (iii) by being more accurate, they can better succeed in multi-step tasks (where compounding errors can cause failure).


**Details**

*Technical quality -* Good. Strengths: The authors show an impressive full-stack implementation and very nice real-world demos. The point cloud-conditioned skill policies are a valuable technical contribution addition to the language-to-action stack. Weaknesses: More could be done to evaluate the ability of the low-level skills on their own, i.e. how well can they generalize to different objects/shapes/poses of objects, but the focus is more on how these skills integrate into the full stack (rather than trying to make the low-level skills perfect) so this is mostly okay.

*Originality -* Good. Strengths: Incorporating the local point cloud for conditioning a local skill policy that outputs 6-DoF "waypoint pose" is a nice addition to existing full-stack language-conditioned manipulation pipelines, and showing a clever way to parameterize it (i.e., by predicting offsets from the local point cloud) is also valuable. Weaknesses: PerAct/CLIPort part is the same as before, SayCan part is the same.

*Clarity -* Good. Strength: The overall pipeline makes sense (high- vs. low-level, language to keypoint + skill, local point cloud to dense waypoint-pose sequence). Weakness: The most critical part of the method to understand (the skill-conditioned waypoint policies) was pretty confusing in terms of some key technical details.
- Details on confusion:
    - My confusion comes from how the local point cloud + mask are used to output the waypoint poses. I cannot tell if it's (1) or (2): (1) The *whole* scene point cloud is used to predict the offset poses, and the predicted mask is appended to this entire point cloud as an extra binary feature, (2) Only the *subset* of the point cloud that comes from the predicted mask is used to predict the offset poses.
    - i.e., in the bowl example that's discussed, you say “this yields a set of ‘candidate’ points for consider for interaction”… and “the bottom of the bowl points predict the points on the rim”, which makes me think you mask out the local point cloud and **only** use these to represent the gripper pose predictions.
    - While this might seem like a minor detail, there are two reasons why I believe it's a critical point to clarify. First - masking + conditioning on the local point cloud alone seems like it would support significantly better generalization. Second - representing the poses as offsets from the local point cloud alone seems like it could make the task more difficult, since the gripper might need to be quite far from these masked points. This has implications for tasks where the gripper is using, e.g., a large tool, such as the broomstick task shown in PerAct (this is mentioned in the "questions for rebuttal" section below).

*Impact/Significance -* Good. Strength: Language-conditioned pipelines are here to stay, so providing more instantiations of how they can work and perform more flexible behaviors is valuable. Further, introducing different kinds of hierarchical decomposition into such pipelines (vs. just end-to-end, direct execution of voxel-predicted actions, etc.) is also beneficial. Weakness (minor): This particular method for low-level skills will probably be built on and improved, but showing the impact of a locally-conditioned component is likely to help expand others’ minds for other ways to do this.


**Other comments, points of confusion**
- In introducing the paper + method, the claimed advantage of “*capacity for long-horizon reasoning”* felt slightly off-putting. IIUC, the main part that enables this is the LLM instruction following, which isn't really a contribution of the work, and the ability to execute individual steps more accurately. This is a contribution of the work, but the claim of enabling "long-horizon reasoning" should then be situated more clearly to this improved ability to perform individual steps more successfully.
- In the related work, this comparison to RL/IL was pretty confusing and hard to unpack: “*Imitation learning (IL) aims to learn skills from demonstrations in a more sample efficient manner than RL, but these skills still fail to generalize to scene perturbations [35, 36]. Furthermore, in both IL and RL, connecting learned skills to precise language in a generalizable fashion is an open challenge. KITE avoids learning skills from scratch and instead defines a library of keypoint-conditioned skills, where the exact parameters of each skill are learned from demonstration.”*

**Quality Of The Limitations Section:**

Limitations are addressed clearly

**Questions For Rebuttal:**

- Address weakness comments above (more clearly explain local skill policies)
- On website example, PerAct fails on "Open the bottom drawer" with a "wrong handle prediction". Why wouldn’t KITE have the same issue?
- How would the waypoint pose policies behave for, i.e., sweeping with a broomstick (as shown in PerAct)?
    - In PerAct, global attention is used to directly predict the next gripper pose (so this can be a function of something far away, like the tip of the broom). In KITE, does representing waypoint poses as offsets from the point cloud limit the ability to handle a task like this (where the "task-relevant region" is far from the gripper, such as if it's holding a tool)?
- How does the method deal with the K different (sequential) waypoints? Is K manually specified for each skill?
- Please improve notation/terminology consistency - you have “keypoint-conditioned skills" and “demonstrations of keypoints/waypoints", but these waypoints are not just points (they are poses)


**Robotics Focus:**

Sufficient demonstration on hardware

**Summary Of Paper:**

This paper aims to perform 6-DoF manipulation from RGB-D + language instructions. The motivation/technical problem: existing instruction-following manipulation pipelines (1) don’t effectively allocate attention toward context at both the scene and object levels and (2) output 6-DoF gripper trajectories with limited expressivity. The proposed pipeline aims to address this: (1) from RGB-D + language, a discrete skill and a region for executing the skill are selected (via keypoint prediction, similar to CLIPort + SayCan), (2) from a local point cloud near the predicted region, a network predicts a dense sequence of "waypoint poses" for the gripper to follow. The method is compared to baselines that map RGB-D + language directly to gripper poses and that use frozen vision-language models for language grounding. The results highlight the benefits of continuous waypoint pose prediction, especially in enabling multi-step tasks, and the use of keypoint-prediction grounding over frozen VLMs.

**Summary Of Recommendation:**

While the details of the main novel technical components should be made more clear, the full-stack language-to-action system implementation is impressive, the addition of hierarchy, local geometry conditioning, and more expressive low-level motion prediction is valuable, and the real-world demos are very well done.

---

### Official Review · Reviewer_nGAo · 2023-07-19

**Confidence:** 4
**Originality:** Good
**Technical Quality:** Good
**Clarity Of Presentation:** Fair
**Impact:** 3

**Recommendation:**

Weak Accept: I recommend accepting the paper, but will not argue for my recommendation if the majority of other reviewers have a different opinion.

**Review:**

I would think that a paper on language understanding would have a much richer set of things the language could refer to.
Each of the three evaluation domains had about 4 skills to talk about, not 400. There seem to be up to 10 or so objects or parts
one could refer to, although this was not clear in the paper.

Why is the success rate in all cases 70-75%? Why is that range so low? And why are rates similar
across very different tasks with "varying tiers of difficulty"? Seems weird.

I am not convinced that selecting regions is more difficult than selecting keypoints, and in many cases I would think regions would
be more useful.
Where does the keypoint go if:
- there are multiple handles on the drawer (many have at least two).
- there is a long graspable area the width of the drawer.
- you have cabinets doors or drawers with no handles, which one pushes to open.
https://www.youtube.com/watch?v=B2xYIQhZloI
Rather than focusing on "keypoints", don't you want to focus on "saliency"?
As the paper says, one is directing attention to particular features, not
commanding contact points.

What is the evidence for "capacity for long-horizon reasoning and generalization."?

Smaller points

"We assume that at least one “grounding” camera can partially see all relevant objects"
This seems unnecessary, and difficult to obtain. Why can't a virtual grounding image or set of images be synthesized instead?

What is the difference between a waypoint policy and a controller to move
between waypoints? Does this detail matter?

In training, why aren't new observations and point clouds obtained at every
point along the robot trajectory, or at least at the waypoints?
Why must a sequence of future waypoints be predicted?


**Quality Of The Limitations Section:**

Limitations are addressed clearly

**Questions For Rebuttal:**

I would suggest have a much more extensive set of skills and objects in the evaluations, and maybe even more evaluation domains.
These could be evaluated in simulation.


**Robotics Focus:**

Sufficient demonstration on hardware

**Summary Of Paper:**

This paper advocates the use of keypoints in 2D images to help disambiguate language for downstream language understanding such
as selecting and parameterizing skills.

**Summary Of Recommendation:**

I am not impressed by the evaluation.

---

### Official Review · Reviewer_pCg6 · 2023-07-20

**Confidence:** 3
**Originality:** Very Good
**Technical Quality:** Good
**Clarity Of Presentation:** Fair
**Impact:** 4

**Recommendation:**

Weak Accept: I recommend accepting the paper, but will not argue for my recommendation if the majority of other reviewers have a different opinion.

**Review:**

Advantages:
- The problem of mapping natural language instructions to robot manipulation is hard, and this paper solves it reasonably well.
- The method offers generalization to unseen tasks (through LLMs) so long as the tasks can be broken down to a set of skills. It also offers generalization to unseen objects (through the use of CLIP and the particular way the keypoint policies are conditioned).
- Overal convincing real-world results showing efficacy of the system compared to prior state of the art methods.

Disadvantages:
- The writing is currently confusing, mixing definitions, inference, learning and data processing in text without sometimes a coherent structure. This makes it hard to keep track of what is the proposed method vs what is auxiliary processing needed to instantiate it in this paper, what are the inputs and outputs of the different components, and what processing happens during data preparation, training, and inference stages. It's not all bad though - all the needed information is there and the paper can be understood sufficiently well, but it doesn't make it easy - I almost need to draw diagrams while reading the paper to make sense of the system.
- Grounding module identifies a single keypoint for instruction and does not appear to be able to ground instructions referring to more than one thing.
- It would be good to compare the against PerAct on RLBench, a more standard and replicable benchmark. Or otherwise provide replicable simulation experiments.
- Given PerAct's good performance before, it is hard to understand why it performs so poorly in these experiments. I suspect it is not used as intended. PerAct is a keypoint policy, so it should be compared in an ablation when used as a keypoint policy within KITE. Comparing KITE to PerAct directly doesn't really make sense, since PerAct is not an instruction-following policy (it takes instruction input only to disambiguate tasks).


**Quality Of The Limitations Section:**

Limitations are addressed clearly

**Questions For Rebuttal:**

Main question I have is regarding comparison to PerAct and the way PerAct has been used in the experiments (see the Review section)

The below paragraphs provide some feedback from my subjective experience reading the paper. I hope it helps the authors re-organize and improve the presentation. These are just suggestions, there are likely other ways to improve presentation.

Section 3.1, starting line 141 it describes inference, saying that "at time t ... we want to know which skill to execute", then "infer a 2D keypoint", then suddently "find a waypoint policy pi" - do we learn a new policy at every timestep? Do we search among a set of policies? What does "find" mean? Does the sentence starting on line 144 refer to learning instead of inference process? That can't be, because on line 149 it says "restart the above process", which means everything above was the same process. Additionally, it is confusing when the inputs and outputs of policies and controllers are defined inline in a paragraph that describes an inference process. It would be much clearer to separate definitions, task formulation, inference process, and learning in separate sections and paragraphs.

I suggest moving Section 3.2 after Section 3.3. It refers to things like "grounding modules" and "acting modules" that have not been clearly defined thus far. Also, it is hard to understand the motivation and formal structure of the data being collected before knowing the inputs and outputs of the modules being trained.

I would say that language grounding refers to converting human-understandable text to a machine-understandable representation for a given system and context. Under this definition, I find it confusing that "Skill selection" is under "Acting Module", but keypoint proposal is under "Grounding module". Both skill selection and keypoint proposal is language grounding.

Section 3.4 again surprisingly mixes learning and inference. "Skill selection" talks about inference (which as I said I think should be under the grounding module), the following paragraph is on "learning" and the one after that on "inference" again. I think moving "Skill selection" to the grounding module will fix one of these issues. The paragraph starting on line 196 internally also mixes learning, inference, and data processing in a confusing way.

Line 215 - if you use an equation, you need to define all symbols in the equation. I think there three options:
1. Define all symbols
2. remove the equation
3. reduce the equation to have less symbols.

Line 184 - typo
Line 35 - is pick and place a benchmark?

**Robotics Focus:**

Sufficient demonstration on hardware

**Summary Of Paper:**

This paper proposes KITE, a method that maps instructions to robot manipulation actions. It consists of two stages: a higher level stage proposes keypoints to act on and skills to execute. Each skill consists of a learned keypoint policy and a controller to generate low-level motion. The experiments show that KITE can solve a number of challenging rearrangement tasks (including openable containers) on a physical robot.

**Summary Of Recommendation:**

My rating is conditioned on improved presentation of the paper

---

### Author Response · Authors · 2023-08-12
**Overall Response to All Reviewers**

To the reviewers, thank you for taking the time to provide valuable and actionable feedback on KITE! We are encouraged to hear that the reviewers agree that KITE’s use of keypoint grounding and skill composition address several reasoning and manipulation challenges of instruction-following with convincing demonstrations on hardware.

Please refer to https://tinyurl.com/kite-rebuttal, which contains all new results for the rebuttal period.

**To summarize, the overarching changes based on the reviewer feedback are detailed below:**
* Additional results, available at https://tinyurl.com/kite-rebuttal:
* Extensive new simulated experiments on semantic grasping across 50+ diverse objects
* Heatmap visualizations in demonstrating KITE’s emergent multimodal reasoning capabilities
* Substantial writing changes in the method section and refined contributions in the introduction/conclusion

**We first address common concerns pointed out by the reviewers here, and provide per-point responses to each reviewer in the sections below:**
* **Presentation improvements:** Reviewers pCg6 and C4rM point out that the current presentation of KITE in the method section (Section 3) would benefit from restructuring. We thank the reviewers for their many useful writing suggestions which we have since incorporated! We fully agree and have substantially re-organized this section with the following major changes: introducing grounding and acting earlier on, clarifying the inputs and outputs to different system components, clarifying the implementation of skills. In the newly uploaded PDF of the manuscript called kite_revisions.pdf, we highlight these changes in blue. We have also modified the introduction and conclusion to focus the contributions of this work around the use of skills locally conditioned on keypoints. As pointed out by Reviewer C4rM, we want to emphasize that these skills (1) give KITE an edge over existing instruction-following frameworks in terms of precision, and (2) enable KITE to “integrate with a [wider range of instructions (referring to objects or object parts)].”
* **Confusion regarding grounding:**
   *  Reviewers pCg6 and nGAo raised an important point regarding how keypoints as a representation handle multimodality – i.e. when different objects or multiple points within a spatial region would satisfy a given instruction, instead of a single point of interaction. To clarify, KITE is in fact capable of multimodal attention. KITE’s grounding module maps an image and current language instruction to a full-resolution heatmap. Since this module is trained from (image, instruction, keypoint) pairs, any multimodality in the demonstration data will be reflected in the distribution of the learned heatmaps.
   *  In our demonstrations, instructions like “Open any drawer” are paired with images where the annotated keypoint lies on the top, middle, or bottom drawer.  Thus, an emergent property of the learned heatmap distributions is that they do attend to salient regions rather than individual points; we ultimately obtain a localized keypoint by argmaxing over this heatmap. We have since added visualizations of this in the additional experiments on the website.


**Citations References throughout Rebuttal:**\
[1] Brohan, Anthony, et al. "Do as i can, not as i say: Grounding language in robotic affordances." CoRL. PMLR, 2023.\
[2] Shridhar, Mohit, et. al. "Perceiver-actor: A multi-task transformer for robotic manipulation." CoRL. PMLR, 2023.\
[3] Parashar, Priyam, et al. "Spatial-Language Attention Policies for Efficient Robot Learning." arXiv preprint arXiv:2304.11235 (2023).\
[4] Brohan, Anthony, et al. "Rt-1: Robotics transformer for real-world control at scale." arXiv preprint arXiv:2212.06817 (2022).\
[5] Stone, Austin, et al. "Open-world object manipulation using pre-trained vision-language models." arXiv preprint arXiv:2303.00905 (2023).\
[6] Goyal, Ankit, et al. "RVT: Robotic View Transformer for 3D Object Manipulation." arXiv preprint arXiv:2306.14896 (2023).

---

> ### Author Response · Authors · 2023-08-12
> **Overall Response to All Reviewers (cont.)**
>
> * **Claims regarding long-horizon reasoning and generalization**
>    * The reviewers suggest that the claims regarding KITE’s long-horizon reasoning and generalization capabilities require further justification. Regarding long-horizon reasoning, we defer to Reviewer C4rM’s observation that “by being more accurate, [KITE] can better succeed in multi-step tasks (where compounding errors can cause failure).” We admit that this is more a consequence of KITE’s precise keypoint-based grounding rather than an explicit design choice. To more explicitly handle the long-horizon case in future work, KITE could also readily integrate with off-the-shelf LLMs, which have been shown to propose reasonable task plans and paired skills for free-form long-horizon instructions [1].
>    * Regarding generalization, we empirically show that KITE is able to generalize across the following axes: 1) unseen object instances, 2) varying degrees of clutter and distractor objects, 3) language variations, and 4) randomized object poses in our real-world experiments. To further illustrate KITE’s generalization, we have since added extensive experiments on semantic grasping of over 50+ simulated objects, now available on the website (https://tinyurl.com/kite-rebuttal).
> * **Baseline evaluation & PerAct reproducibility**
>    * Reviewers pCg6 and C4rM seek to better understand KITE’s reported performance gains over PerAct. We posit that the following hardware and implementation differences in our setup, as compared to original PerAct, could impact performance.
>    * ***Demonstration Collection:*** First, the original PerAct’s demonstrations were collected using VR teleoperation in which entire trajectories are recorded. In post-processing, keyframes (voxelized observation, action) are extracted from these trajectories using various heuristics (i.e. joint velocities are zero, gripper state changed). Critically, this heuristic post-processing step is a cheap way to inflate tens of demonstration trajectories to several hundred state-action pairs with which to train the transformer. In our setting, we use kinesthetic teaching instead. We find this to be more amenable to recording sparse waypoints, as required to instantiate KITE’s skills. To create a PerAct-compatible dataset within our setup, this means we must manually start and stop the robot’s motion before recording state-action pairs, to avoid the human demonstrator being in view.
>    * ***Training Procedure:*** For a controlled comparison, we train KITE and PerAct on the same data volume (50 state-action pairs per skill) and for comparable train times (a few hours). Originally, PerAct was trained on a much larger data volume, due to augmentations made available by heuristic filtering, and for 16 days.
>    * ***Hardware Differences:*** Additionally, original PerAct used a single Kinect sensor for RGB-D observations. In our setup, we have 3 RealSense cameras available, but these are known to yield noisier depth along with possible artifacts of merged point clouds. In KITE, the precision of local keypoints in an RGB image helps to mitigate this. PerAct instead deals with large receptive field over the global voxel grid, where point cloud noise can certainly lead to imprecision in our reimplementation.
>    * We note that prior works [3] which reproduce PerAct similarly observe 0-40% PerAct success in the real world on comparable tasks to ours (Tables 2 and 3). [3] also leverages kinesthetic teaching and controls for the same data volume during training of PerAct and their proposed approach. Lastly, we note that success rate alone is a fairly unforgiving measure of task progress. To best understand PerAct’s performance, we kindly refer the reviewers to videos on our website (https://tinyurl.com/kite-manip)  to see its reasonable behavior but struggle with precision.

---

### Decision · Program_Chairs · 2023-08-30

**Decision:**

Accept (Poster)

**Comment:**

### Summary, Strengths and Weaknesses
This paper introduces KITE, a two-stage method that utilizes keypoints to map instructions to robot manipulation actions. Experimental results demonstrate that KITE outperforms models that use visual language or end-to-end control. It achieves high success rates due to precise keypoint-driven execution in real-world environments. While the formulation of the method needs further refinement, the method is novel. Real-world demonstrations showcase the effectiveness of the approach.

The reviewers agree in their recommendation to accept the paper because most of the concerns have been addressed. I agree with their consensus.